# Using a pragmatically adapted, low-cost contingency management intervention to promote heroin abstinence in individuals undergoing treatment for heroin use disorder in UK drug services (PRAISE): a cluster randomised trial

Nicola Metrebian ,[1] Tim Weaver,[2] Kimberley Goldsmith,[3] Stephen Pilling,[4] Jennifer Hellier,[3] Andrew Pickles,[3] James Shearer,[5] Sarah Byford,[5] Luke Mitcheson,[6] Prun Bijral,[7] Nadine Bogdan,[8] Owen Bowden-Jones,[9] Edward Day,[10] John Dunn,[11] Anthony Glasper,[12] Emily Finch,[6] Sam Forshall,[13] Shabana Akhtar,[10] Jalpa Bajaria,[8] Carmel Bennett,[10] Elizabeth Bishop,[4] Vikki Charles,[1] Clare Davey,[13] Roopal Desai,[1] Claire Goodfellow,[4] Farjana Haque,[1] Nicholas Little,[4] Hortencia McKechnie,[1,14] Franziska Mosler,[1] Jo Morris,[13] Julian Mutz,[1] Ruth Pauli,[10] Dilkushi Poovendran,[14] Elizabeth Phillips,[8] John Strang,[1,6] on behalf of the Contingency Management Programme Team

NM and TW are joint first authors.

For numbered affiliations see end of article.

**Correspondence to**
Dr Nicola Metrebian;
nicola.metrebian@kcl.ac.uk

## ABSTRACT

**Introduction** Most individuals treated for heroin use disorder receive opioid agonist treatment (OAT)(methadone or buprenorphine). However, OAT is associated with high attrition and persistent, occasional heroin use. There is some evidence for the effectiveness of contingency management (CM), a behavioural intervention involving modest financial incentives, in encouraging drug abstinence when applied adjunctively with OAT. UK drug services have a minimal track record of applying CM and limited resources to implement it. We assessed a CM intervention pragmatically adapted for ease of implementation in UK drug services to promote heroin abstinence among individuals receiving OAT.

**Design** Cluster randomised controlled trial.

**Setting and participants** 552 adults with heroin use disorder (target 660) enrolled from 34 clusters (drug treatment clinics) in England between November 2012 and October 2015.

**Interventions** Clusters were randomly allocated 1:1:1 to OAT plus 12× weekly appointments with: (1) CM targeted at opiate abstinence at appointments (CM Abstinence); (2) CM targeted at on-time attendance at appointments (CM Attendance); or (3) no CM (treatment as usual; TAU). Modifications included monitoring behaviour weekly and fixed incentives schedule.

**Measurements** Primary outcome: heroin abstinence measured by heroin-free urines (weeks 9–12). Secondary outcomes: heroin abstinence 12 weeks after discontinuation of CM (weeks 21–24); attendance; self-reported drug use, physical and mental health.

**Strengths and limitations of this study**

► To our knowledge, this is the first clinical trial to examine the effectiveness of contingency management (CM) in promoting heroin abstinence among those in opioid agonist treatment in the UK.

► This study is a large cluster randomised trial conducted at 34 drug treatment services in England.

► CM, as developed in the USA, was pragmatically adapted for ease of implementation in resource-poor UK substance use treatment settings.

► Adaptions included using drug service staff to deliver all aspects of the CM intervention (as opposed to CM specialists); staff were trained, supported by a CM Handbook and regular supervision.

► Illicit drug use was monitored through urine drug screens once a week rather than more frequently using a fixed rather than escalating schedule.

**Results** CM Attendance was superior to TAU in encouraging heroin abstinence. Odds of a heroin-negative urine in weeks 9–12 was statistically significantly greater in CM Attendance compared with TAU (OR=2.1; 95% CI 1.1 to 3.9; p=0.030). CM Abstinence was not superior to TAU (OR=1.6; 95% CI 0.9 to 3.0; p=0.146) or CM Attendance (OR=1.3; 95% CI 0.7 to 2.4; p=0.438) (not statistically significant differences). Reductions in heroin use were not sustained at 21–24 weeks. No differences between groups in self-reported heroin use.

**Conclusions** A pragmatically adapted CM intervention for routine use in UK drug services was moderately effective in encouraging heroin abstinence compared with no CM only when targeted at attendance. CM targeted at abstinence was not effective.

**Trial registration number** ISRCTN 01591254.

## INTRODUCTION

Opioid agonist treatment (OAT) (methadone and buprenorphine) is recognised globally as a clinically[1][2] and cost-effective[2][3] treatment for opiate use disorder. However, its effectiveness is often undermined by high attrition associated with relapse into illicit drug use.[4] In the UK, the National Institute for Health and Care Excellence recommends psychological therapies (including contingency management (CM)) be offered alongside OAT[2] to support behaviour change.

CM is a behavioural intervention based on the principles of operant conditioning,[5] delivered as a time-limited adjunct to existing evidence-based treatments (such as OAT) to amplify patient benefit. It involves providing positive reinforcement (usually monetary vouchers or prizes) contingent on achieving prespecified behaviour consistent with treatment goals.[6][7]

Evidence, primarily from US trials, shows that CM is an effective adjunct to substance use treatment in encouraging abstinence from drug use, including in treating drug use regardless of treatment setting, and for treating drug use (including cocaine, opiates and cocaine and polysubstance use) in opiate addiction treatment.[8–12] However, there is weaker evidence for the effectiveness of CM in encouraging abstinence from illicit opiates among those receiving OAT.[9][12–14]

UK guidance, based on the evidence at the time,[8][9] recommended CM should be used in UK drug treatment to target the reduction of illicit drug use and encourage attendance at appointments.[2][15] However, UK drug treatment services had little experience of applying CM, reduced treatment budgets and limited staff capacity; conditions likely to impede implementation of CM.[6] We aimed to assess a CM intervention pragmatically adapted for ease of implementation in routine UK practice. We have previously demonstrated that low-cost CM can affect short-term behaviour change of improved adherence to a hepatitis B vaccination schedule[16] achieving clear long-term health economic benefit.[17] The trial reported here focused on more complex and longer term behaviour change. Specifically, we assessed the effectiveness of two different 12-week CM schedules targeting (1) opiate abstinence or (2) attendance at clinical appointments, providing immediate positive reinforcement through vouchers delivered by trained staff, in achieving heroin abstinence among individuals receiving OAT in community drug treatment settings.

## METHODS
### Study design and setting

We employed a pragmatic cluster randomised controlled trial design described elsewhere.[18] The unit of randomisation was the drug clinic (cluster) rather than individual participant. This was for three reasons: (1) to reduce the likelihood of contamination if staff were delivering and patients receiving different interventions at the same drug clinic; (2) to reduce interpatient contamination as patients themselves constitute a local social network; and (3) to reduce the risk of low recruitment, poor compliance and high dropout within treatment as usual (TAU) arm if participants receiving TAU were denied an incentive offered to others in the same clinic. Sites were recruited in stages and then randomised. Thirty-four drug treatment clinics (clusters) in England (National Health Service (NHS): London, Birmingham, Sussex, Essex, Bath and Bristol, Dudley and Walsall; non-NHS: London, Hertfordshire and Birmingham) were randomly allocated to one of three conditions (described below) and tasked with recruiting a cluster sample of 22 participants. Clinics were eligible if they provided OAT (methadone or buprenorphine), weekly clinical appointments and received enough OAT referrals to meet recruitment targets.

Within each cluster, participants received the same allocated condition, thus minimising the risk of contamination between intervention and control arms.

The trial tested the following research hypotheses:

▶ CM (positive reinforcement targeted at treatment attendance) will increase abstinence from street heroin when compared with TAU in which no positive reinforcement is offered.

▶ CM (positive reinforcement targeted at the provision of opiate-negative urine samples at treatment appointments) will increase abstinence from street heroin when compared with TAU in which no positive reinforcement is offered.

▶ Differences in the type of CM schedules will be associated with differences in heroin abstinence.

Having both CM interventions allowed us to assess whether any benefit (abstinence) derives from a direct effect of CM on abstinence as the behavioural target, or is a general benefit resulting from CM-stimulated improved attendance at clinical appointments (and possibly consequent improvements in treatment retention).

Findings from parallel economic and process evaluations are to be reported separately.

### Participants

Eligible patients were aged 18 and above; seeking a new episode of OAT; regular users of street heroin (ie, self-reported use 15/preceding 30 days (at least 3 days/week), and all (minimum 1) urine drug screens (UDS) in previous month positive for opiates); meeting International Classification of Diseases 10th Revision criteria for opiate dependence; willing to receive 12-week CM intervention; at liberty to participate in the study for 24 weeks; and willing and able to provide informed consent. We excluded patients if they could not read English and required an interpreter to understand a brief oral description of the study to ensure they would be able to

understand the CM intervention provided; were pregnant or breast feeding (due to being seen as a special population receiving special treatment provision); and/or were referred through the criminal justice pathway and were receiving a community sentence on condition of attending drug treatment as they would be subject to additional contingencies which might influence their behaviour.

## Randomisation and masking

Each cluster (drug clinic) was randomly assigned to OAT plus 12× weekly clinical appointments with either (1) CM Abstinence: positive reinforcement contingent on opiate abstinence monitored through UDS undertaken at each weekly appointment; (2) CM Attendance: positive reinforcement contingent on attendance on time; or (3) no CM (TAU; control condition).

Randomisation was undertaken independently by the King's Clinical Trials Unit. Clusters were assigned to treatments using random permuted blocks within type of service provider strata (NHS or non-NHS) using a block length of 3 in a 1:1:1 allocation ratio. Laboratory personnel who completed the urinalysis and the statistician analysing primary outcome data were all blinded to treatment allocation.

## Interventions

OAT was delivered in line with existing service protocols at all clinics and included weekly clinical appointments between 15 and 50 min with a named drug worker. Attending the service to obtain a prescription did not constitute attendance. The CM interventions (described below) were delivered adjunctively.

In our trial, CM, as developed in the USA, was pragmatically adapted for use in resource-poor UK drug treatment services to ease future implementation. Key CM principles were retained; targeting a clearly defined behaviour, regular monitoring of that behaviour and providing an immediate reinforcer ensured that a clear contingent relationship was made between the positive behaviour and the reinforcer. The reinforcer was withheld if the target behaviour was not achieved. Adaptions included: (A) training drug workers to deliver CM (as opposed to CM specialists); (B) monitoring illicit drug use through weekly UDS (rather than more frequently) to fit routine UK practice of taking UDS once a week at the start of treatment and less frequently thereafter; and (C) using a fixed rather than escalating schedule, as drug service staff had previously reported the use of escalating schedules to be too difficult.[19] CM consisted of positive reinforcement—verbal praise and fixed value of £10 supermarket voucher at each appointment. Reasoning that anything lower would be unlikely to encourage participants to attend while escalation of the value would increase the costs of the intervention beyond most services budget.

'*CM Abstinence*': Reinforcement was conditional on attendance at weekly appointments and (A) during weeks 1–4 (priming weeks) providing a UDS whatever the result

and (B) during weeks 5–12 providing an opiate-negative UDS. UDS were undertaken using a drug integrated cup test to detect opioids. This test is able to detect opioids up to 1–3 days after use. While unable to confirm opioid abstinence over the week period, the test nevertheless provided clinically significant evidence of the participants' ability to abstain from using opioids over this briefer period. Instant UDS tests were completed at the start of each appointment. Staff provided the reinforcer immediately the target behaviour was achieved. Priming ensured participants gained experience of the reinforcer, while also addressing practitioner concerns about the difficulty in achieving abstinence during contemporaneous titration up to a stable dose.

'*CM Attendance*': Eligibility to receive the reinforcement was conditional solely on on-time attendance at the appointment (within 15 min of the scheduled appointment time).

Control: Participants received TAU (OAT plus 12× weekly appointments) with no CM.

The CM interventions ceased after week 12, at which point the frequency of appointments reverted to usual care for each service (ie, varied depending on drug service and patient needs).

## Training and supervision

All staff providing clinical appointments received training on trial procedures. At CM sites, staff received bespoke 1-day training on the principles and practice of CM (including simulation, role-play). Training was supported by a CM Handbook designed for UK services but which drew on international evidence and practice models.[20]

CM was delivered in the first 5–10 min of the appointment and audio recorded. A CM adherence measure was developed from measures available at the time[21] but adapted to assess key CM competencies in UK service settings. Audio recordings were used to rate the scale which comprised 10 four-point Likert scale items. Adherence was scored as poor (<33%), adequate (33%–66%) or good (>66%). The scale achieved good inter-rater reliability ($k=0.63$; $p<0.001$).

Evidence suggests training alone is unlikely to change health professional behaviour unless supported by effective supervision.[22] Group supervision sessions were provided to staff at intervention sites throughout the trial. The drug service psychologist/senior practitioner provided local supervision after receiving training in CM supervision. Audio recordings of CM delivery were available to local supervisors to provide feedback to staff. In turn, research team psychologists used recordings of supervision sessions to support their supervision of local supervisors.

## Research assessments

Consenting participants completed a research interview (and were reimbursed £20 for time and travel) before enrolment into the trial and again at 12 and 24 weeks after enrolment. All participants were asked to provide

(research) UDS weekly between weeks 9–12 and 21–24. Weekly 'urine collection clinics' were established by the research team at each clinic (participants were reimbursed for travel) to enable participants to provide UDS outside of appointments. Participants' attendance at appointments, receipt of CM vouchers and provision of urines for CM and research purposes were recorded.

## Outcomes

The primary outcome measure of heroin abstinence was a once-weekly binary positive/negative heroin UDS collected at weeks 9–12 for laboratory drug testing.

Secondary outcome measures included retention in treatment, self-reported illicit drug use in the last 30 days,[23] alcohol use (Alcohol Use Disorders Identification Test (AUDIT)),[24] social functioning (Opiate Treatment Index (OTI)),[25] physical and mental health status (Short Form-36 (SF-36))[26] and depression and anxiety (Hospital Anxiety and Depression Scale),[27] obtained by researchers in face-to-face interviews with patients at baseline, and 12 and 24 weeks. Confidential patient ratings of therapeutic alliance with their health professional were recorded at 4, 8 and 12 weeks after enrolment (Agnew Relationship Measure-5).[28] A single-item measure of delay discounting[29] and a motivation measure for drug abstinence[30] were ancillary outcome measures and reported elsewhere.

The Adult Service Use Schedule adapted for drug users,[31] health-related quality of life (EuroQoL 5-Dimension 3-Level)[32] and the WHO Health and Work Performance Questionnaire[33] were measured for health economic analysis. Findings from the health economic analysis and parallel process evaluation are to be reported separately.

## Sample size calculation

More detail is provided in the protocol.[18] Lussier *et al*[8] estimated a mean weighted effect size of 0.39 from three studies with an opiate use outcome measure (CM vs control). For a two-sided test, alpha=0.05, 5% attrition, 80% power to detect an effect size of 0.39, 111 participants per group were required. Accounting for clustering by clinic, this was inflated to 220 participants per arm (assuming an intraclass correlation of 0.05, 20 participants per cluster and 11 clusters per intervention leading to a design effect of 1.95). Clinics were initially advised to recruit >20 participants each (33×20=660 in total). After recruiting 13 clusters, with an attrition rate of 10%, larger than the expected 5%, clusters still recruiting were asked to increase recruitment from 20 to 22 participants.

## Statistical analysis

Statistical analysis was carried out using Stata V.14/15 according to the intention-to-treat principle. Variables were summarised using mean/SD, median/quartiles or frequencies/proportions as appropriate.

Refusal and non-attendance UDS were coded as positive (positive assumption) but results missing for other reasons (eg, discharged from treatment, in hospital or in prison) were initially left missing. This positive assumption is commonly used in clinical practice and in studies of CM and was used in our previous trial.[34]

These responses were the dependent variables in a mixed effects logistic regression model (9–12 and 21–24 weeks in separate models), with trial arm, the clinic type stratification factor and week dummy variables as fixed effects, and random intercepts for participants nested within clinic. Results are presented both from fitting a model fitted to data where (1) the positive assumption has been applied, but no further imputation has taken place, (2) further missing data were imputed (100 data sets) and estimates obtained using Rubin's rules. The imputation model included clinic, trial arm, NHS stratification factor, gender, heroin use at 12 and 24 weeks, and baseline measures including frequency of heroin use, AUDIT score, OTI social functioning score anxiety and depression, SF-36 physical and mental component scores and delay discounting, in addition to the four binary weekly urine sample measures. Moderation by type of opioid replacement (methadone or buprenorphine), NHS clinic versus non-NHS and the Charlson Index of multiple drug use was tested using the non-imputed data by adding treatment by moderator interaction terms each separately to the model described.

Differences in secondary continuous outcomes between the groups were estimated using linear mixed effects models with the 12 and 24-week outcomes as dependent variables, the same fixed and random effects as for the primary outcome, plus time point, treatment by time interaction and baseline values of the outcome where applicable as predictors. For the self-report binary drug use variables, marginal logistic generalised estimating equation (GEE) regression models were fitted (to estimate marginal instead of potentially inflated conditional effects from mixed models, which were also fitted to compare),[35] with the same variables as the linear mixed models and with clinic as a clustering variable. Missing data were not imputed.

Retention in the 12-week treatment programme was coded as weeks attended up to the point of continuous future non-attendance. Retention was compared between the groups using: (1) Fisher's exact test for retention in the trial overall, (2) a logistic GEE model for binary retention in opioid treatment at 12 weeks (with exchangeable correlation structure and 1000 repetition bootstrap SEs), and (3) Kaplan-Meier and Cox regression analyses for time until continuous treatment non-attendance, with trial group and NHS stratification factor as predictors and clinic as a clustering variable.

## Patient and public involvement

A Service User Research Advisory Group (SURAG) was established before the trial commenced, comprising 14 service users with direct experience of receiving OAT. The group met once every 6 months to provide advice on the design, conduct and progress of the research. The

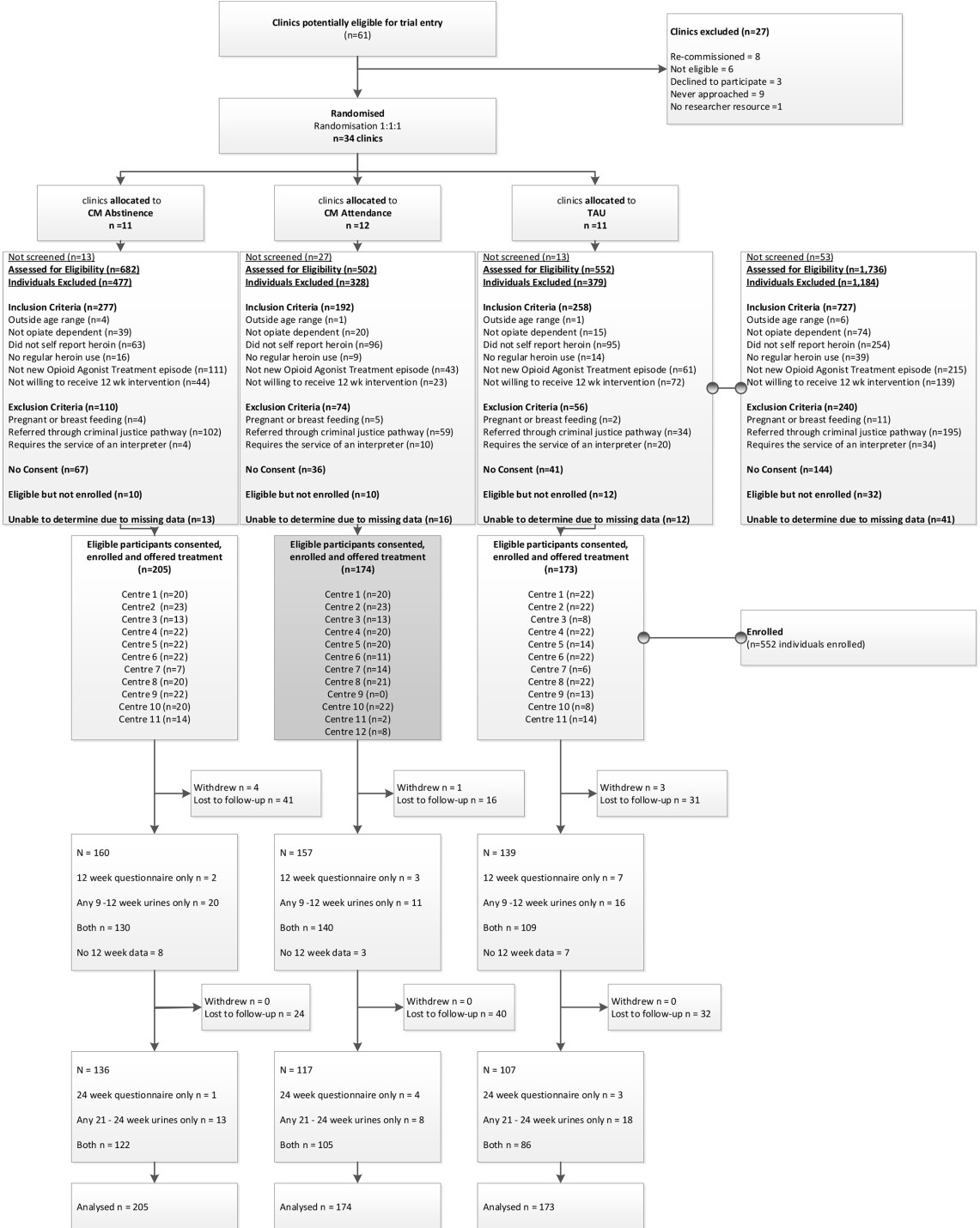

**Figure 1** Consolidated Standards of Reporting Trials (CONSORT) diagram. CM, contingency management; TAU, treatment as usual.

SURAG helped significantly to make the trial procedures acceptable and aided implementation.

## RESULTS
### Sample
Between September 2012 and September 2015, we randomly allocated 34 clinics (clusters) (of 61 considered) to three treatments (figure 1; and see online supplemental material). Of the 34 clinics, all provided OAT, the majority were NHS (62%) and just under one-half were within London (47%). Figure 1 shows that across 34 clusters, 789 patients were screened for eligibility, of whom 552 (31.8%) were consented and enrolled (between November 2012 and October 2015), a figure that fell short of our target of 660. One cluster did not recruit any participants. Although we recruited the target number of sites, we were unable to recruit 20 (and latterly 22) participants at all sites. Recruiting sites and participants at each site

were challenging due to retendering of service contracts which affected many sites during the trial period. Two services were decommissioned during the trial.

Retention in the trial was high and comparable across trial arms. Eight participants actively withdrew from the trial (CM Abstinence=4 (2%); CM Attendance=1 (1%); TAU=3 (2%), p=0.50).

## Characteristics

Study participants were broadly representative of patients entering OAT in England (table 1). They were mostly male (404; 73%), white (435; 79%) and with a mean age of 38.2 years (SD 8.8). They had first used opiates at the mean age of 23 years (SD 8), first injected at a mean age of 26 years (SD 8) and first received treatment at a mean age of 30 years (SD 8). They had previously been in opiate treatment a median of two times (25th, 75th percentiles 1 to 4). Over half had been in prison (291; 53%). Fifty-seven per cent were prescribed methadone (315), 43% buprenorphine (237).

All participants (552) self-reported using heroin in the month prior to interview (including two reporting using non-prescribed pharmaceutical heroin). Three-quarters reported using crack (419; 76%) while 23% used benzodiazepines (n=129). Half drank alcohol (53%; n=310) and nearly all used tobacco (531; 96%) (table 2; and see online supplemental material).

There was reasonable balance across the three trial arms in terms of baseline sociodemographic, drug use, health and treatment experience variables. However, there were more women in the CM Attendance group, fewer white individuals in the CM Abstinence group and a smaller proportion with insecure housing status in the CM Attendance group.

## Abstinence from heroin at 12 weeks

Of the 2208 scheduled UDS (552 participants × 4 urines for weeks 9–12), 601 (27%) were not obtained due to non-compliance (refusal or DNA) and were imputed as positive (positive assumption), with the greatest number of these being in the TAU group. Additionally, 454 (21%) were treated as missing, including: 260 (12%) not obtained due to unable to contact, 77 (3%) not obtained due to clinic error, 15 (1%) unable to provide a sample, 55 (3%) in prison (47) and in hospital (8), and for 10 (0.5%) the reason was missing. The percentage of UDS results missing by week and trial arm are shown in table 3.

Results from the analyses with and without imputation were similar (table 4), so results from imputed data are discussed. Participants in the CM Attendance group had statistically significantly greater odds of a heroin-negative urine at 9–12 weeks compared with those in the TAU group (OR=2.1; 95% CI 1.1 to 3.9; p=0.030). There were no statistically significant differences between the CM Abstinence and TAU group (OR=1.6; 95% CI 0.9 to 3.0; p=0.146) or CM Attendance group (OR=1.3; 95% CI 0.7 to 2.4; p=0.438).

Figure 2 shows the observed (dashed line) and model-predicted (solid line) expected probabilities of a heroin-negative urine over the 4 weeks of the primary outcome period. Based on estimated proportions from model 1 in table 4, a higher proportion of those in CM Attendance (62%) or CM Abstinence (57%) provided a negative heroin urine sample at week 12 compared with TAU (48%). The intraclass correlations from the non-imputed data were 0.03 (95% CI 0.01 to 0.11) at the clinic level.

There was no moderation of trial intervention outcome by medication or treatment setting (buprenorphine vs methadone ($\chi^2(2)$=1.89; p=0.389); NHS or non-NHS service ($\chi^2(2)$=4.96; p=0.084)) or by other or multiple drug use (Charlson Index of multiple drug use ($\chi^2(2)$=4.93; p=0.085)).

## Abstinence from heroin at 24 weeks

There were no statistically significant differences among groups in heroin abstinence for weeks 21–24 (table 4).

Figure 3 shows the observed (dashed line) and model 1 expected probabilities (solid line) of a heroin-negative UDS over the 4 weeks of the follow-up period (weeks 21–24). The estimated proportions of those providing a negative heroin UDS at 24 weeks from model 1 in figure 3 were similar across groups: CM Abstinence (44%); CM Attendance (45%); TAU (37%).

## Attendance

Figure 4 shows that 60% of those receiving CM Abstinence and 59% receiving CM Attendance attended their first appointment on time, compared with 46% of the TAU group. Eleven per cent of participants did not attend any weekly appointments (CM Abstinence=11%; CM Attendance=7%; TAU=14%).

Attrition at each week was highest in the TAU group, with the proportion of participants attending appointments falling from 46% to 24% between weeks 1 and 12. In the CM Abstinence group, there was a decline in attendance from 53% (week 4) down to 45% (week 5) with a steady decline thereafter to 33% attending at week 12. Attendance in the CM Attendance group remained consistent with a slight decline from 59% (week 1) to 51% (week 12). Most attendance was on time (within 15 min) (figure 4).

## Retention in 12 weekly appointments

The proportions attending all of the 12-week appointments differed across the groups, with a higher proportion attending in CM Attendance as compared with CM Abstinence and TAU groups (56% vs 39% and 30%). Participants in the CM Attendance group had statistically significantly greater odds of full attendance compared with those in TAU (OR=3.1; 95% CI 2.0 to 4.6; p<0.001). There were no statistically significant differences between CM Abstinence and TAU (OR=1.5; 95% CI 0.9 to 2.4; p=0.099).

Participants in the CM Attendance group attended longer than those in the CM Abstinence group or TAU

**Table 1** Baseline characteristics by treatment group

| | | CM Abstinence | | | CM Attendance | | | TAU | | | Total | | |
|---|---|---|---|---|---|---|---|---|---|---|---|---|---|
| | | n | Mean or median (%) | SD or IQR | n | Mean or median (%) | SD or IQR | n | Mean or median (%) | SD or IQR | n | Mean or median (%) | SD or IQR |
| **Demographics** | | | | | | | | | | | | | |
| Age at baseline (years), mean | | 205 | 38.39 | 8.76 | 174 | 38.02 | 8.48 | 173 | 38.08 | 9.30 | 552 | 38.18 | 8.83 |
| Gender | Male | 160 | 78.05 | | 118 | 67.82 | | 126 | 72.83 | | 404 | 73.19 | |
| | Female | 45 | 21.95 | | 56 | 32.18 | | 47 | 27.17 | | 148 | 26.81 | |
| Ethnicity | White | 145 | 70.73 | | 145 | 83.33 | | 145 | 83.82 | | 435 | 78.80 | |
| | Black | 15 | 7.32 | | 5 | 2.87 | | 13 | 7.51 | | 33 | 5.98 | |
| | Asian | 23 | 11.22 | | 11 | 6.32 | | 8 | 4.62 | | 42 | 7.61 | |
| | Mixed | 18 | 8.78 | | 13 | 7.47 | | 6 | 3.47 | | 37 | 6.70 | |
| | Not done | 4 | 1.95 | | 0 | 0 | | 1 | 0.58 | | 5 | 0.91 | |
| Employment | Employed | 13 | 6.34 | | 27 | 15.52 | | 22 | 12.72 | | 62 | 11.23 | |
| | Unemployed/sickness | 190 | 92.68 | | 146 | 83.91 | | 145 | 83.82 | | 481 | 87.14 | |
| | Student | 0 | 0 | | 0 | 0 | | 1 | 0.58 | | 1 | 0.18 | |
| | Housewife/husband | 2 | 0.98 | | 0 | 0 | | 1 | 0.58 | | 3 | 0.54 | |
| | Retired | 0 | 0 | | 0 | 0 | | 2 | 1.16 | | 2 | 0.36 | |
| | Other | 0 | 0 | | 1 | 0.57 | | 2 | 1.16 | | 3 | 0.54 | |
| Normally live with | Partner/spouse | 33 | 16.10 | | 43 | 24.71 | | 27 | 15.61 | | 103 | 18.66 | |
| | Friends | 16 | 7.80 | | 11 | 6.32 | | 14 | 8.09 | | 41 | 7.43 | |
| | Alone | 89 | 43.41 | | 75 | 43.10 | | 82 | 47.40 | | 246 | 44.57 | |
| | Family | 45 | 21.95 | | 31 | 17.82 | | 31 | 17.92 | | 107 | 19.38 | |
| | Other | 21 | 10.24 | | 14 | 8.05 | | 19 | 10.98 | | 54 | 9.78 | |
| | Unknown | 1 | 0.49 | | 0 | 0.00 | | 0 | 0.00 | | 1 | 0.18 | |
| Accommodation | Owner occupier | 22 | 10.73 | | 13 | 7.47 | | 8 | 4.62 | | 43 | 7.79 | |
| | Rented private | 33 | 16.10 | | 44 | 25.29 | | 34 | 19.65 | | 111 | 20.11 | |
| | Rented (LA, HA) | 75 | 36.59 | | 75 | 43.10 | | 73 | 42.20 | | 223 | 40.40 | |
| | Living with parent | 23 | 11.22 | | 15 | 8.62 | | 14 | 8.09 | | 52 | 9.42 | |
| | B&B/hotel | 2 | 0.98 | | 3 | 1.72 | | 2 | 1.16 | | 7 | 1.27 | |
| | Hostel | 28 | 13.66 | | 12 | 6.90 | | 10 | 5.78 | | 50 | 9.06 | |
| | NFA (living on the streets) | 21 | 10.24 | | 12 | 6.90 | | 31 | 17.92 | | 64 | 11.59 | |
| | Other | 0 | 0 | | 0 | 0 | | 1 | 0.58 | | 1 | 0.18 | |
| | Unknown | 1 | 0.49 | | 0 | 0 | | 0 | 0 | | 1 | 0.18 | |
| Prison | No | 95 | 47.03 | | 80 | 46.24 | | 81 | 47.09 | | 256 | 46.80 | |

Continued

**Table 1** Continued

| | | CM Abstinence | | | CM Attendance | | | TAU | | | Total | | |
|---|---|---|---|---|---|---|---|---|---|---|---|---|---|
| | | n | Mean or median (%) | SD or IQR | n | Mean or median (%) | SD or IQR | n | Mean or median (%) | SD or IQR | n | Mean or median (%) | SD or IQR |
| | Yes | 107 | 52.97 | | 93 | 53.76 | | 91 | 52.91 | | 291 | 53.20 | |
| Times on remand | | 105 | 6.28 | 8.69 | 92 | 5.27 | 5.59 | 88 | 5.10 | 7.59 | 285 | 5.59 | 7.45 |
| **Drug use history** | | | | | | | | | | | | | |
| Age first used opiates (years) | | 200 | 23.39 | 8.01 | 169 | 23.67 | 7.65 | 173 | 23.35 | 7.48 | 542 | 23.46 | 7.72 |
| Age first regular use (years) | | 200 | 25.30 | 8.18 | 173 | 26.02 | 7.90 | 173 | 26.28 | 7.92 | 546 | 25.84 | 8.01 |
| Age first injected (years) | | 121 | 26.10 | 7.85 | 92 | 26.51 | 8.48 | 108 | 26.32 | 7.46 | 321 | 26.29 | 7.88 |
| Age first received help (years) | | 193 | 30.23 | 8.36 | 170 | 29.45 | 7.56 | 167 | 30.00 | 8.54 | 530 | 29.91 | 8.16 |
| **Drug treatment** | | | | | | | | | | | | | |
| Treatment | Methadone | 119 | 58.05 | | 106 | 60.92 | | 90 | 52.02 | | 315 | 57.07 | |
| | Buprenorphine (sub) | 86 | 41.95 | | 68 | 39.08 | | 83 | 47.98 | | 237 | 42.93 | |
| Treatment dose (mean mg) | Methadone | 119 | 34.45 | 13.05 | 103 | 31.41 | 7.49 | 90 | 34.32 | 11.90 | 312 | 33.41 | 11.20 |
| | Buprenorphine (sub) | 85 | 6.87 | 4.67 | 68 | 5.90 | 3.15 | 81 | 6.56 | 3.20 | 234 | 6.48 | 3.80 |
| Median times in opiate treatment | | 198 | 2 | 1–4 | 172 | 2 | 1–3 | 171 | 2 | 1–5 | 541 | 2 | 1–4 |

CM, contingency management; LA, HA, Local Authority, Housing Association; TAU, treatment as usual.

**Table 2** Opiate Treatment Index (OTI) measures of heroin use in the last 30 days measured at baseline, and 12 and 24 weeks

| | CM Abstinence n=205 | CM Attendance n=174 | TAU n=173 | Total n=552 |
|---|---|---|---|---|
| **Summary statistics for binary Yes/ No to 'use in the past 30 days'?** (Missing shown but n (%) proportions calculated disregarding missing) | n (%) | n (%) | n (%) | n (%) |
| **Baseline** | | | | |
| Used heroin in the last 30 days | 205 (100) | 173 (99) | 172 (99) | 550 (99.6) |
| Did not use heroin in the last 30 days | 0 (0) | 1 (1) | 1 (1) | 2 (0.4) |
| *Missing* | 0 | 0 | 0 | 0 |
| **12 weeks** | | | | |
| Used heroin in the last 30 days | 96 (73) | 111 (78) | 84 (72) | 291 (74) |
| Did not use heroin in the last 30 days | 36 (27) | 32 (22) | 32 (28) | 100 (26) |
| *Missing* | 73 | 31 | 57 | 161 |
| **24 weeks** | | | | |
| Used heroin in the last 30 days | 97 (79) | 85 (78) | 66 (75) | 248 (78) |
| Did not use heroin in the last 30 days | 26 (21) | 24 (22) | 22 (25) | 72 (23) |
| *Missing* | 82 | 65 | 85 | 232 |
| **Number of days taking heroin in the last 30 days** | | | | |
| **Baseline** | | | | |
| n | 204 | 173 | 173 | 550 |
| Number missing | 1 | 1 | 0 | 2 |
| Proportion missing | 0.49 | 0.57 | 0 | 0.36 |
| Median (25th, 75th percentiles) | 28.00 (22.00, 30.00) | 27.00 (21.00, 30.00) | 28.00 (21.00, 30.00) | 28.00 (21.00, 30.00) |
| **12 weeks** | | | | |
| n | 132 | 143 | 116 | 391 |
| Number missing | 73 | 31 | 57 | 161 |
| Proportion missing | 35.61 | 17.82 | 32.95 | 29.17 |
| Median (25th, 75th percentiles) | 4.00 (0.00, 15.00) | 5.00 (1.00, 13.00) | 4.00 (0.00, 15.00) | 4.00 (0.00, 15.00) |
| **24 weeks** | | | | |
| n | 123 | 108 | 88 | 319 |
| Number missing | 82 | 66 | 85 | 233 |
| Proportion missing | 40.00 | 37.93 | 49.13 | 42.21 |
| Median (25th, 75th percentiles) | 5.00 (1.00, 15.00) | 4.50 (1.00, 15.00) | 3.50 (0.50, 20.00) | 4.00 (1.00, 15.00) |

CM, contingency management; TAU, treatment as usual.

group. Participants in the CM Abstinence group had statistically significantly greater risk of dropping out of the appointments before week 12 compared with those in TAU group (HR=1.9; 95% CI 1.5 to 2.5; p<0.001) and in CM Attendance group (HR=1.7; 95% CI 1.2 to 2.3; p=0.002) (figure 5).

## Clinical outcomes: self-reported drug use

Self-reported heroin use reduced in all groups from nearly 100% (99.6%; 550) to 74% (291) at week 12, while the median number of days used reduced from 28 (21, 30) to 4 (0, 15). Self-reported use of crack and benzodiazepines reduced in the trial population from 76% to 54% and 23% to 16%, respectively, at 12 weeks. Self-reported alcohol and tobacco use remained similar (from 56% to 61% and 96% to 95%, respectively). There were no significant differences in these outcomes between groups at 12 weeks (tables 2 and 5; and see online supplemental material).

At 24-week follow-up, we found no difference in self-reported illicit drug use between groups. However,

**Table 3** Urine results—heroin (using opiates and 6-MAM variables in combination+recoding missing urines to positive if participant DNA or refused to provide=positive assumption)

| | Week 9 | | | | Week 10 | | | | Week 11 | | | | Week 12 | | | |
|---|---|---|---|---|---|---|---|---|---|---|---|---|---|---|---|---|
| | CM Abstinence | CM Attendance | TAU | Total | CM Abstinence | CM Attendance | TAU | Total | CM Abstinence | CM Attendance | TAU | Total | CM Abstinence | CM Attendance | TAU | Total |
| Negative | 73 / 35.61 | 58 / 33.33 | 44 / 25.43 | 175 / 31.7 | 59 / 28.78 | 67 / 38.51 | 44 / 25.43 | 170 / 30.8 | 70 / 34.15 | 78 / 44.83 | 51 / 29.48 | 199 / 36.05 | 82 / 40 | 95 / 54.6 | 62 / 35.84 | 239 / 43.3 |
| Positive | 77 / 37.56 | 78 / 44.83 | 86 / 49.71 | 241 / 43.66 | 96 / 46.83 | 71 / 40.8 | 85 / 49.13 | 252 / 45.65 | 80 / 39.02 | 67 / 38.51 | 83 / 47.98 | 230 / 41.67 | 69 / 33.66 | 56 / 32.18 | 73 / 42.2 | 198 / 35.87 |
| Missing | 55 / 26.83 | 38 / 21.84 | 43 / 24.86 | 136 / 24.64 | 50 / 24.39 | 36 / 20.69 | 44 / 25.43 | 130 / 23.55 | 55 / 26.83 | 29 / 16.67 | 39 / 22.54 | 123 / 22.28 | 54 / 26.34 | 23 / 13.22 | 38 / 21.97 | 115 / 20.83 |
| Total | 205 | 174 | 173 | 552 | 205 | 174 | 173 | 552 | 205 | 174 | 173 | 552 | 205 | 174 | 173 | 552 |

| | Week 21 | | | | Week 22 | | | | Week 23 | | | | Week 24 | | | |
|---|---|---|---|---|---|---|---|---|---|---|---|---|---|---|---|---|
| | CM Abstinence | CM Attendance | TAU | Total | CM Abstinence | CM Attendance | TAU | Total | CM Abstinence | CM Attendance | TAU | Total | CM Abstinence | CM Attendance | TAU | Total |
| Negative | 63 / 30.73 | 68 / 39.08 | 45 / 26.01 | 176 / 31.88 | 51 / 24.88 | 36 / 20.69 | 30 / 17.34 | 117 / 21.2 | 47 / 22.93 | 40 / 22.99 | 36 / 20.81 | 123 / 22.28 | 56 / 27.32 | 52 / 29.89 | 49 / 28.32 | 157 / 28.44 |
| Positive | 75 / 36.59 | 56 / 32.18 | 73 / 42.2 | 204 / 36.96 | 87 / 42.44 | 80 / 45.98 | 87 / 50.29 | 254 / 46.01 | 86 / 41.95 | 81 / 46.55 | 78 / 45.09 | 245 / 44.38 | 75 / 36.59 | 72 / 41.38 | 66 / 38.15 | 213 / 38.59 |
| Missing | 67 / 32.68 | 50 / 28.74 | 55 / 31.79 | 172 / 31.16 | 67 / 32.68 | 58 / 33.33 | 56 / 32.37 | 181 / 32.79 | 72 / 35.12 | 53 / 30.46 | 59 / 34.1 | 184 / 33.33 | 74 / 36.1 | 50 / 28.74 | 58 / 33.53 | 182 / 32.97 |
| Total | 205 | 174 | 173 | 552 | 205 | 174 | 173 | 552 | 205 | 174 | 173 | 552 | 205 | 174 | 173 | 552 |

CM, contingency management; 6-MAM, 6-monoacetylmorphine; TAU, treatment as usual.

**Table 4** Primary outcome analysis: odds of subjects providing a heroin-negative urine over weeks 9–12 and 21–24

| | Contrast | OR | 95% CI | P value |
|---|---|---|---|---|
| **Weeks 9–12** | | | | |
| (1) Outcome: repeated binary heroin result with refused and DNA=positive sample (n=483, no multiple imputation) | CM Abstinence versus TAU | 1.67 | 0.89 to 3.14 | 0.114 |
| | CM Attendance versus TAU* | 2.19 | 1.15 to 4.15 | 0.017 |
| | CM Attendance versus CM Abstinence | 1.31 | 0.71 to 2.44 | 0.391 |
| (2) Outcome: repeated binary heroin result with refused and DNA=positive sample (ie, analysis 1) and remaining missing multiply imputed (n=552, using multiple imputation) | CM Abstinence versus TAU | 1.59 | 0.85 to 3.01 | 0.146 |
| | CM Attendance versus TAU* | 2.05 | 1.07 to 3.91 | 0.030 |
| | CM Attendance versus CM Abstinence | 1.29 | 0.68 to 2.41 | 0.438 |
| **Weeks 21–24** | | | | |
| (1) Repeated binary analysis for weeks 21–24 with refused and DNA=0 (n=409, no multiple imputation) | CM Abstinence versus TAU | 1.71 | 0.72 to 4.09 | 0.225 |
| | CM Attendance versus TAU | 1.66 | 0.69 to 4.00 | 0.261 |
| | CM Attendance versus CM Abstinence | 0.97 | 0.41 to 2.28 | 0.939 |
| (2) Analysis 1 with remaining missing multiply imputed (n=552, using multiple imputation) | CM Abstinence versus TAU | 1.46 | 0.66 to 3.26 | 0.352 |
| | CM Attendance versus TAU | 1.29 | 0.57 to 2.92 | 0.543 |
| | CM Attendance versus CM Abstinence | 0.89 | 0.40 to 1.93 | 0.751 |

*Significant at 0.05.
CM, contingency management; TAU, treatment as usual.

there was weak evidence of increased odds of alcohol use in the last 30 days in CM Attendance as compared with CM Abstinence (OR=1.9; 95% CI 1.0 to 3.5; p=0.048), and evidence of fewer days using tobacco in the last 30 days in TAU as compared with CM Abstinence (OR=2.1; 95% CI −0.3 to −3.9; p=0.024) (see online supplemental material). Mixed effects models gave similar results.

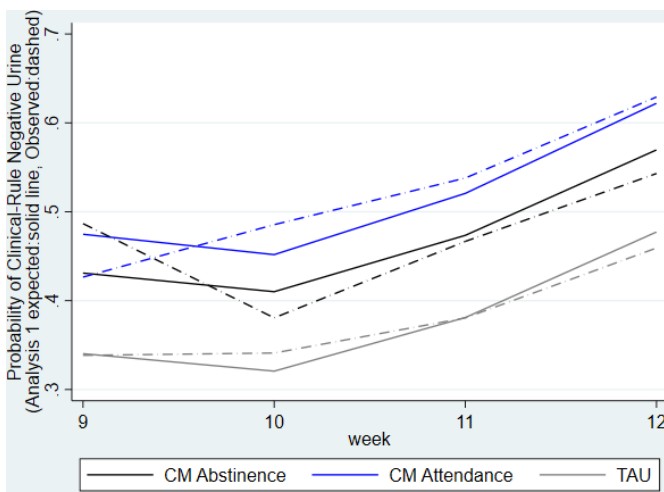

**Figure 2** Probability of heroin-negative urine applying positive assumption over weeks 9–12. CM, contingency management; TAU, treatment as usual.

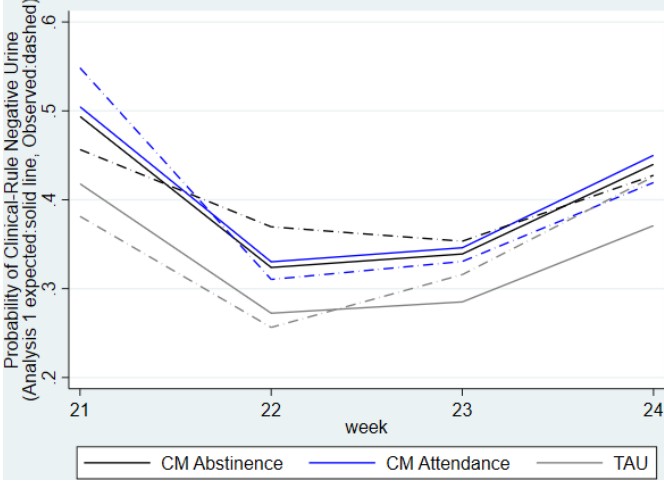

**Figure 3** Probability of heroin-negative urine applying positive assumption over weeks 21–24. CM, contingency management; TAU, treatment as usual.

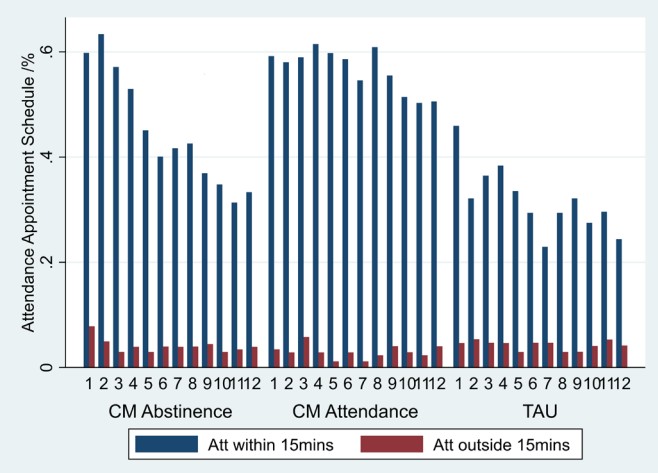

**Figure 4** Attendance at appointments. CM, contingency management; TAU, treatment as usual.

## Clinical outcomes: health and social

There were no differences between the groups in clinical outcomes at week 12 or 24 (for statistical tests see online supplemental material). A similar therapeutic alliance score was reported by all groups at 4, 8 and 12 weeks (see online supplemental material).

## Adherence to CM

Both CM Abstinence and CM Attendance achieved a score of 'adequate' adherence to the CM manual. It is important to note these averages were on the lower end of the range of scores in this category (CM Abstinence=38.33%, SD=9.49 (range 24.6%–56.3%); CM Attendance=44.15%, SD=17.07 (range 16.7%–68.2%)).

## Serious adverse events

There were 25 serious adverse events in 22 participants: CM Abstinence=10 in 9 participants; CM Attendance=8 in 7 participants; TAU=7 in 6 participants. The number of deaths was 3 (CM Abstinence=1; CM Attendance=2). None were related to the trial intervention.

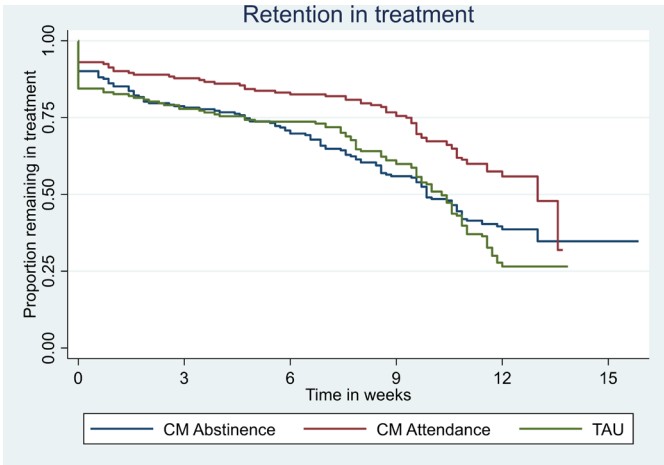

**Figure 5** Time to continuous appointment non-attendance. CM, contingency management; TAU, treatment as usual.

## DISCUSSION

We found the adapted CM intervention targeted at attendance was moderately effective in encouraging heroin abstinence compared with TAU. CM Abstinence was not more effective than TAU or CM Attendance.

The UK's clinical guidance,[2 15] which promotes the use of CM, was influenced by systematic reviews and meta-analysis conducted at least 10 years ago which showed CM was effective in encouraging abstinence from a range of drugs.[9] Previous reviews examining the effectiveness of CM in promoting opioid abstinence have been mixed[9–12 14] with the most recent systematic review and meta-analysis concluding there was no evidence of CM working better than control in encouraging abstinence from non-prescribed opiates during OAT.[14] Our findings concur with the most recent meta-analysis; although adapting CM for ease of implementation in resource-poor settings may have caused it to lose some of its potential effectiveness.

When our adapted CM intervention was targeted at attendance only, it was superior to TAU in encouraging heroin abstinence. It has been argued that clinical benefit arises from regular attendance at clinical appointments and overall retention in treatment.[36] Our findings suggest CM can improve attendance (when targeted at it) among populations who often prove challenging to engage with and retain in treatment and that the clinical benefit we observed supports such arguments.

The CM interventions we evaluated retained the main principles of CM, but modifications were made to ensure ease of implementation in UK settings with limited treatment budgets.

First, in the UK, resources are limited. Patients receiving OAT attend their service and receive UDS once a week at the start of treatment and less frequently thereafter. Previous studies exploring views of CM among UK drug treatment staff and patients found that staff viewed the US frequency of urine testing to be too resource heavy for the UK settings and service users felt strongly that such a regime would act as a disincentive.[37] Hence, opiate abstinence was monitored through once-weekly UDS rather than multiple tests per week. Opioids can only be detected up to 1–3 days after use so the UDS cannot confirm opioid abstinence over the week period, only provide evidence of abstinence over this briefer period, potentially reducing the impact of the reinforcer. As CM Abstinence had no effect on either UDS or self-reported heroin use, it is possible that CM Abstinence only encouraged abstinence in the 1–3 days before an appointment and a UDS test. While it may be argued that such behaviour change can be clinically significant to the individual patient, the absence of change in secondary outcomes suggests we should be cautious about inferring any significant clinical benefit.

Second, while the CM evidence base demonstrates the efficacy of escalating value incentives,[8–13] we used a fixed-value incentive. In a previous UK study of CM targeted at hepatitis B vaccination, undertaken by the authors, we

**Table 5** ORs for Opiate Treatment Index (OTI) measures of heroin use in the last 30 days assessed at 12 and 24 weeks: use in the last 30 days (binary) and number of days used

| | Treatment group difference estimate | Lower confidence limit | Upper confidence limit | P value |
|---|---|---|---|---|
| Heroin use in the last 30 days Y/N across groups—GEE model results | | | | |
| 12-week CM Attendance versus CM Abstinence | 1.28 | 0.77 | 2.14 | 0.336 |
| 12-week TAU versus CM Abstinence | 0.98 | 0.56 | 1.70 | 0.936 |
| 12-week TAU versus CM Attendance | 0.76 | 0.47 | 1.23 | 0.268 |
| 24-week CM Attendance versus CM Abstinence | 0.95 | 0.51 | 1.78 | 0.876 |
| 24-week TAU versus CM Abstinence | 0.80 | 0.40 | 1.63 | 0.545 |
| 24-week TAU versus CM Attendance | 0.85 | 0.42 | 1.69 | 0.633 |
| Mean differences number of days used in the last 30 days across groups | | | | |
| 12-week CM Attendance versus CM Abstinence | −0.42 | −3.11 | 2.27 | 0.760 |
| 12-week TAU versus CM Abstinence | 0.02 | −2.77 | 2.80 | 0.990 |
| 12-week TAU versus CM Attendance | 0.44 | −2.33 | 3.21 | 0.757 |
| 24-week CM Attendance versus CM Abstinence | −0.25 | −3.06 | 2.56 | 0.859 |
| 24-week TAU versus CM Abstinence | −0.13 | −3.06 | 2.80 | 0.931 |
| 24-week TAU versus CM Attendance | 0.13 | −2.85 | 3.10 | 0.934 |

CM, contingency management; GEE, generalised estimating equation; TAU, treatment as usual.

found escalating schedule was more difficult to implement and for staff to adhere to.[16] Employing escalating schedules might have risked unacceptable levels of protocol violations while also increasing the costs of the intervention, and limiting potential for translation into routine clinical practice. We used £10 vouchers—a figure our SURAG regarded as sufficient to encourage participants to attend and provide a UDS.

Third, it was critical that the CM schedule was acceptable to participants and supported by staff. Our SURAG argued that having to provide opiate-negative UDS would be challenging during titration and staff felt monitoring at this point may be undermining. In response, we introduced an initial 4-week priming phase in the CM Abstinence arm, during which participants received reinforcement for on-time provision of a UDS, irrespective of the result. The use of priming in CM ensures that patients experience receipt of the reinforcer early in treatment.[6] As participants were entering OAT and undergoing dose titration for their methadone/buprenorphine medication, priming provided an opportunity for participants to receive exposure to the incentive without the expectation that they needed to be heroin abstinent during this period.

Fourth, vouchers were chosen as reinforcers as there is good evidence for their effectiveness[11 12] and costs were relatively low given the once-a-week frequency of reinforcement.

Adherence to CM as set out in the training manual was somewhat better in the CM Attendance group than CM Abstinence group but not significantly so. Movement of trained staff and supervisors away from participating services may well have impacted on adherence. Further exploration of the relationship between adherence and outcomes and the factors which may affect this (such as individual competence or service organisational factors) will help to clarify this.

Although CM has a strong evidence base in achieving behaviour change, there is weaker evidence for behaviours sustained once the incentives are removed. Previous studies have found differing results with only a small treatment effect following discontinuation of incentives.[11] In this study, the reduction in heroin use was not sustained after discontinuation of CM. We are unable to conclude whether this was due to the cessation of CM or weekly appointments. These appointments were not routine practice in most drug services after 12 weeks but considered good clinical practice in the first 12 weeks of treatment.[38] Further research is needed to better understand how to maintain positive behaviour change, including whether the reinforcer needs to be maintained, given intermittently, or tapered. We also need to explore the impact of the cessation, or reduced frequency of appointments after 12 weeks.

While there is a growing body of evidence for CM, it has not been widely implemented in UK drug treatment services. Similarly, the actual adoption and implementation of CM programmes in US community-based treatment programmes (including those where most research on CM has been undertaken) is rare,[39] possibly due to limited staff capacity, resources and treatment budgets[6] though there are some reports of successful implementation.[40]

Adapting CM for ease of implementation in resource-poor settings may have led to a loss of some of its potential effectiveness. Therefore, future research should consider how best to implement CM in routine practice both in UK and international contexts and should consider

different models of delivery (including the use of mobile telephone technology).

**Author affiliations**
[1]Addictions, King's College London, Institute of Psychiatry, Psychology and Neuroscience, London, UK
[2]Department of Mental Health & Social Work, Middlesex University, London, UK
[3]Biostatistics and Health Informatics, King's College London, Institute of Psychiatry, Psychology and Neuroscience, London, UK
[4]Centre for Outcomes, Research and Effectiveness, University College London, London, UK
[5]Health Services and Population Research, King's College London, Institute of Psychiatry, Psychology and Neuroscience, London, UK
[6]Addictions, South London and Maudsley NHS Foundation Trust, London, UK
[7]Management Offices, Change Grow Live, Manchester, UK
[8]Sankey House, Essex Partnership University NHS Foundation Trust, Pitsea,Essex, UK
[9]Addictions and Substance Misuse, Central and North West London NHS Foundation Trust, London, UK
[10]Addiction Services, Birmingham and Solihull Mental Health NHS Foundation Trust, Birmingham, UK
[11]Drugs and Alcohol Services, Camden and Islington NHS Foundation Trust, London, UK
[12]Substancce Misuse Service, Sussex Partnership NHS Foundation Trust, Worthing, UK
[13]Drug and Alcohol Services, Avon and Wiltshire Mental Health Partnership NHS Trust, Bath, UK
[14]Centre for Mental Health, Imperial College London, London, UK

**Acknowledgements** We would like to thank all the drug service staff and service users who participated in this trial. We also gratefully acknowledge the support and guidance of members of our Trial Steering Committee (Simon Coulton (chair), Christopher Whiteley and Soraya Mayet); our Data Management and Ethics Committee (Louise Sell (chair), Anne R Lingford-Hughes and Zoe Hoare); and our Service User Research Advisory Group and the late Anthea Martin for facilitating this group. Also, we would like to acknowledge the guidance and support we received from the late Professor Nancy Petry who collaborated on this trial.

**Contributors** The corresponding author had full access to all the data in the study and had final responsibility for the decision to submit for publication. Concept, original design (JS, TW, SP, SB). Trial administration and management (NM, TW). Treatment development, training and supervision (SP, LM). Statistical analysis (KG, JH, AP, JS, SB). Implementation lead medical clinicians (PB, NB, OBJ, ED, JD, AG, EF, SF). Data management (NM, FH). Data handling and entry (SA, JB, CB, EB, VC, CD, RD, CG, FH, NL, HMK, FM, JMo, JMu, RP, DP, EP). Data interpretation (NM, TW, KG, JS). First draft manuscript (NM, KG, TW). Further work on manuscript (NM, TW, KG, SP, JS, SB, JSh). All authors read and approved the final manuscript.

**Funding** This paper presents independent research funded by the National Institute for Health Research (NIHR) under its Programme Grants for Applied Research Programme (grant reference number RP-PG-0707-10149). The research was also part funded by the NIHR Biomedical Research Centre at South London and Maudsley NHS Foundation Trust and King's College London.

**Disclaimer** The views expressed are those of the authors and not necessarily those of the National Health Service, the NIHR or the Department of Health.

**Competing interests** JS, SP and LM have contributed to UK guidelines on the potential role of contingency management in the management of opioid addiction (NICE, 2007; convened by SP, chaired by JS). SP receives funding from NICE for the production of clinical guidelines. JS has chaired the broader scope pan-UK working group preparing the 2017 and 2007 Orange Guidelines for the UK Departments of Health and Social Care, providing guidance on management and treatment of drug dependence and misuse, including guidance on possible inclusion of contingency management. LM and ED contributed to these guidelines. JS is a researcher and clinician who, through his university, has worked with various pharmaceutical companies to identify new or improved treatments and his employer (King's College London) has received grants, travel costs and/or consultancy payments from companies including, past 3 years, Indivior, Mundipharma, Camurus, Molteni Farma and Accord. JS has also worked with various drug policy organisations and advisory bodies including the Society for the Study of Addiction (SSA) and the European Monitoring Centre for Drugs and Drug Addiction (EMCDDA). JS and KG are supported by the National Institute for Health Research (NIHR) Biomedical Research Centre for Mental Health at South London and Maudsley NHS Foundation Trust and King's College London. JS is an NIHR senior investigator. For a fuller account, see JS's web page at http://www.kcl.ac.uk/ioppn/depts/addictions/people/hod.aspx. NM is involved in research project funded by pharmaceutical company Mundipharma. LM has been in receipt of an untied educational grant from Indivior for the ARC study which incorporated the use of CM in the intervention arm. LM is currently involved in pharmaceutical company (Indivior) funded study–the EXPO trial which will incorporate the use of CM as part of the psychosocial intervention. LM has a paid secondment to PHE to advise on best practice psychological interventions in drug and alcohol treatment. ED has recently been appointed as the government's Drug Recovery Champion.

**Patient consent for publication** Not required.

**Ethics approval** The Positive Reinforcement targeting Abstinence In Substance misusE (PRAISE) trial was reviewed at the NRES Committee South East Coast-Surrey (12/LO/0910) and received a favourable opinion on 25 July 2012.

**Provenance and peer review** Not commissioned; externally peer reviewed.

**Data availability statement** Data are available upon reasonable request. Data collected for the study, including individual participant data and a data dictionary defining each field in the set, will be made available to others. These data will include deidentified participant data, data dictionary, study protocol, statistical analysis plan and informed consent form. These data will be available with publication and on application to NM. Access criteria for obtaining data will include approval from the chief investigator (JS), approval of study proposal and with a signed data access agreement.

**ORCID iDs**
Nicola Metrebian http://orcid.org/0000-0003-3581-1703
Sarah Byford http://orcid.org/0000-0001-7084-1495
Claire Goodfellow http://orcid.org/0000-0001-5990-8150
Julian Mutz http://orcid.org/0000-0001-5308-1957

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
