## [Reviewer comments · BMJ Open]

ARTICLE DETAILS

TITLE (PROVISIONAL)	Using a pragmatically adapted, low-cost, contingency management intervention to promote heroin abstinence in individuals undergoing treatment for heroin use disorder in UK drug services (PRAISE): a cluster randomised trial
AUTHORS	Metrebian, Nicola; Weaver, Tim; Goldsmith, Kimberley; Pilling, Stephen; Hellier, Jennifer; Pickles, Andrew; Shearer, James; Byford, Sarah; Mitcheson, Luke; Bijral, Prun; Bogdan, Nadine; Bowden-Jones, Owen; Day, Edward; Dunn, John; Glasper, Anthony; Finch, Emily; Forshall, Sam; Akhtar, Shabana; Bajaria, Jalpa; Bennett, Carmel; Bishop, Elizabeth; Charles, Vikki; Davey, Clare; Desai, Roopal; Goodfellow, Claire; Haque, Farjana; Little, Nicholas; McKechnie, Hortencia; Mosler, Franziska; Morris, Jo; Mutz, J; Pauli, Ruth; Poovendran, Dilkushi; Phillips, Elizabeth; Strang, John

VERSION 1 – REVIEW

REVIEWER	Mao, Limin University of New South Wales, Centre for Social Research in Health
REVIEW RETURNED	27-Nov-2020

GENERAL COMMENTS	This paper presents a standard cluster RCT across 34 sites (clusters) across UK to examine whether an add-on contingency management intervention was superior among patients receiving ongoing opioid pharmacotherapy. It is a well-designed, well-written and logically presented paper. I have two minor comments for the authors to consider in the discussion section:  1. some comments about retention of the cohort where the majority has ongoing socioeconomic disadvantages (e.g., unemployed, history of crime). 2. reflections on implementation of CM-abstinence and CM-(timely) attendance across different sites-whether there were any cultural changes, site variations.
--

REVIEWER	Dr Munyaradzi Madhombiro The University of Zimbabwe, Zimbabwe
REVIEW RETURNED	09-Dec-2020

GENERAL COMMENTS	The authors present a well written manuscript and all round attention to detail. The subject is highly significant and the endeavour is worthwhile. While the authors report some effect of
---

	CM on opioid use disorders, they highlight the challenges in implementing the intervention in real life situation in the UK. Please find my review. 1. Outcomes (Item 6a) Completely defined pre-specified primary outcome measure including how and when it was assessed Is it clear (1) what the primary outcome is (usually the one used in the sample size calculation), (2) how it was measured (if relevant; e.g. which score used), (3) at what time point, and (4) what the analysis metric was (e.g. change from baseline, final value)? My comment; The primary outcomes of UDS positive/negative is well described. The secondary outcomes are also described. 2. Sample size (Item 7a) How sample size was determined Is there a clear description of how the sample size was determined, including (1) the estimated outcomes in each group; (2) the α (type I) error level; (3) the statistical power (or the β (type II) error level); and (4) for continuous outcomes, the standard deviation of the measurements? My comment; the sample size determination is well described. 3. Sequence generation (Item 8a) Method used to generate random allocation sequence Does the description make it clear if the “assigned intervention is determined by a chance process and cannot be predicted”? My comment; Sequence generation was well described and appears adequate. 4. Allocation concealment (Item 9) Mechanism used to implement random allocation sequence (such as sequentially numbered containers), describing any steps taken to conceal the sequence until interventions were assigned Is it clear how the care provider enrolling participants was made ignorant of the next assignment in the sequence (different from blinding)? Possible methods can rely on centralised or “third-party” assignment (i.e., use of a central telephone randomisation system, automated assignment system, sealed containers). My comment; This is described well 5. Blinding (Item 11a) If done, who was blinded after assignment to interventions (for example, participants, care providers, those assessing outcomes) Is it clear if (1) healthcare providers, (2) patients, and (3) outcome assessors are blinded to the intervention? General terms such as “double-blind” without further specifications should be avoided.
--	--

	My comment; Participants and healthcare providers were not blinded although the laboratory and statistician were blinded. As this is a cluster design, blinding was probably adequate. 6. Outcomes and estimation (Item 17a/b) For the primary outcome, results for each group, and the estimated effect size and its precision (such as 95% confidence intervals) Is the estimated effect size and its precision (such as standard deviation or 95% confidence intervals) for each treatment arm reported? When the primary outcome is binary, both the relative effect (risk ratio, relative risk) or odds ratio) and the absolute effect (risk difference) should be reported with confidence intervals. My comments; Although retention in care is a secondary outcome, considerable effort is invested in it and blurs the importance of the primary outcome which is the UDS. The reporting should thus emphasize the primary outcome. The authors present the Odds ratios, and that is appropriate, however, for the ease of readers, it is better to report the odds ratio in a standard way such as 2.1(95% CI 1.1 to 3.9;p=0.030) instead of '2.1 times the odds of a heroin-negative urine (95% CI 1.1 to 3.9;p=0.030)'. The self-report outcomes are given as percentages, but without their 95% CI. They would have been better presented with their 95%CI. The authors claim 'CM Abstinence had 1.5 times the odds of attending fully as compared to TAU (CI 0.9 to 2.4;p=0.099)' but this crosses the point of no effect, which is 1. So how significant was this therefore. Even the p=0.099? This needs to be stated as such. The authors did indicate they were not going to present the secondary outcome of economic evaluation which they did not which is fair. 7. Harms (Items 19) All important harms or unintended effects in each group Is the number of affected persons in each group, the severity grade (if relevant) and the absolute risk (e.g. frequency of incidence) reported? Are the number of serious, life threatening events and deaths reported? If no adverse event occurred this should be clearly stated. My comments; This is explained well. 8. Registration (Item 23) Registration number and name of trial registry Is the registry and the registration number reported? If the trial was not registered, it should be explained why. My comments; This is covered well. 9. Protocol (Item 24) Where trial protocol can be accessed Is it stated where the trial protocol can be assessed (e.g. published, supplementary file, repository, directly from author, confidential and therefore not available)?
--	--

	My comments; This is covered. 10. Funding (Item 25) Sources of funding and other support (such as supply of drugs) and role of funders. Are (1) the funding sources, and (2) the role of the funder(s) described? My comments; This is covered well.
--	---

REVIEWER	Mélissa Beaudoin (1) University of Montreal, Canada (2) McGill University, Canada (3) Research center of the Institut universitaire en santé mentale de Montréal, Canada
REVIEW RETURNED	12-Dec-2020

GENERAL COMMENTS	This paper reports the findings of a cluster randomized trial comparing the effects of contingency management (CM), targeting either abstinence or attendance, to treatment as usual in patients with heroin use disorder. CM was pragmatically adapted in order to facilitate its implementation in the UK. The authors concluded that these interventions were moderately effective to encourage heroin abstinence. This study reports very interesting and relevant results for the adoption of future policies in the UK. One of the limitations of CM is the difficulty in implementing it, and therefore this article attempts to address this problem. Although the clinical trial appears to have been well conducted and the reported results are very comprehensive, the authors could better address the study's many limitations and be more careful in interpreting the results. ABSTRACT 1. The term "opioid use disorder" is repeatedly used to describe heroin users across the manuscript. While all heroin users have an OUD, not all people with OUD use heroin. Therefore, I suggest the use of "heroin use disorder" instead. 2. Reading the abstract, it is unclear to me who are the participants exactly (i.e., adults with heroin use disorder). This should be specified. 3. It would be relevant to enumerate a few secondary outcomes as well. 4. Across the manuscript, nonsignificant results are presented similarly to statistically significant results. In order to avoid misleading the reader, it would be important to be careful in the manner of presenting these results. For example, the authors could mention that this result is not significant, or even use the conditional tense (e.g., "CM Abstinence seemed to be superior to TAU, with an odds ratio of...). 5. The conclusions should be more nuanced since (a) the reductions were not sustained in time, (b) the study has many important limitations, and (c) no improvement was observed regarding self-reported heroin use. INTRODUCTION 6. The introduction is overall very concise and clear. However, it would be interesting if the authors provided more detail about the available evidence on CM efficacy in drug, opioid and heroin users.
--

METHODS

7. The study design is well described. However, the rationale justifying the utilization of clusters was not detailed in the article. Nevertheless, it was well explained in the previously published protocol. In order to meet the CONSORT 2010 item 2a, the authors should provide a bit more detail about that in the “Study design and setting” section.

8. It would also be relevant to explain what the rationale is for the choice of the exclusion criteria.

9. If the information is available, for replicability purposes, it would be relevant to specify what the approximative duration of the weekly clinical appointments was.

10. A major concern I have is about the UDS test that was used. What test was used exactly? How sensitive/specific is it? For how long following consumption can it detect heroin in urine? Unless I am mistaken, heroin can no longer be detected in urine after two days. Therefore, a weekly negative sample does not ensure that the patient is abstinent. While this design was chosen to facilitate the ease of implementation, which seems reasonable, the limitations underlying this measurement should nevertheless be explicitly detailed.

11. Regarding the changes that were made to the protocol during the trial (first paragraph of page 7): please provide the rationale for these changes (CONSORT 2010 item 3b).

RESULTS

12. Please refer to comment #4, which is relevant for all the results section.

13. More information should be provided as to why the initial objective of 660 participants was not met.

14. Figure 1: Why nine clinics were never approached?

15. Table 1: There seem to be some differences between the groups. Statistical tests should be conducted to ensure that participants in the three interventions are similar, and statistical adjustments should be made if differences are found.

16. Page 8 lines 51–52: if the two remaining participants were also using non-prescribed pharmaceutical heroin, this means that every single was using heroin in the month preceding the interview? The second sentence seems to be in contradiction with the first one. Please clarify.

17. Figure 3: It would be easier to follow and more coherent with the other figures if the x-axis was presenting weeks instead of days.

18. The number of tables could be significantly reduced and it would be easier to interpret if the statistical results were presented with the associated descriptive statistics (e.g., table 2 with table 3, supp. table 3 with table 4 & 5, etc.).

19. Heroin, crack, benzodiazepine, alcohol and cigarette use are reported. What about the use of other types of opioid? What is the rationale for not including them?

20. Supp. table 2 seems to be the same as table 5. If this is the case, it should be removed.

21. Whenever the authors state that there is “no difference” between the groups (e.g., page 13 line 22), a statistical test should accompany that claim.

22. Page 13 line 39: The authors claim that the serious adverse events were not related to the trial intervention. How was that determined? What was considered to be a serious adverse event, specifically?

	DISCUSSION 23. Please refer to comment #10 which should also be addressed in this section. 24. An important concern I have is that no effect was observed on self-reported heroin use. Considering that heroin use may not always be detected with a weekly UDS, this raises a major limitation. Indeed, the question is whether it is possible that the only effect of CM on some patients is that they did not consume in the 1–2 days before their visit but were still consuming afterward. This important issue should be discussed in that section. 25. In general, the authors should discuss more the limitations of the study and remain careful in their interpretations of the results, especially considering that these are not supported by the secondary outcomes (heroin use, functioning, wellbeing, depression, anxiety). In summary, the research question is relevant, and the study appears to have been conducted adequately. The weaknesses are mainly found in the justification of the choices that were made, in the interpretation of the results and in the recognition of the limitations of the study. If it suits you, I will be happy to receive your answers as well as your corrected manuscript.
--	---

VERSION 1 – AUTHOR RESPONSE

Reviewer: 1

We appreciate the reviewer’s positive comments about design, writing and presentation.

Discussion

The reviewers states, I have two minor comments for the authors to consider in the discussion section:

1. some comments about retention of the cohort where the majority has ongoing socioeconomic disadvantages (e.g., unemployed, history of crime).

We thank the reviewer for this comment and agree that this is worthy of note. In response we have made a small addition to the discussion (which for context should be viewed alongside our response to reviewer 3) as follows: “Our findings suggest CM can improve attendance (when targeted at it) amongst population who often prove challenging to engage with and retain in treatment ...”. We trust we have understood the reviewers intention correctly by interpreting this to be an invitation to celebrate the retention in treatment of a population experiencing socioeconomic disadvantage. We have been cautious in doing so in order not to over-state the significance of this secondary outcome. As other reviewers note there is a need for general caution about interpreting patient benefit when the effect on our primary outcome is modest and improvements in secondary outcomes are absent.”

2. reflections on implementation of CM-abstinence and CM-(timely) attendance across different sites-whether there were any cultural changes, site variations.

We are grateful to the reviewer for posing an interesting and important question. Mindful of the importance of investigating the relationship between treatment process and outcome, and the potential for variability between sites, we incorporated a qualitative process evaluation into the study design. This will be reported separately in a forthcoming paper.”

Reviewer: 2

We appreciate the reviewer's positive comments about the importance of the subject, presentation and detail and their clear assessment of our methods and reporting.

The reviewer stated, Although retention in care is a secondary outcome, considerable effort is invested in it and blurs the importance of the primary outcome which is the UDS. The reporting should thus emphasize the primary outcome.

We thank the reviewer for their comment. We agree we have allocated considerable space in the manuscript to reporting retention and attendance at appointments and have reported this before the primary outcome. We feel we need to report retention and attendance (not least because this was the target behaviour that was reinforced in one trial arm) but have moved it, as suggested after the primary outcome. Please note that the Figures have now been re-numbered.

The reviewer states, The authors present the Odds ratios, and that is appropriate, however, for the ease of readers, it is better to report the odds ratio in a standard way such as 2.1(95% CI 1.1 to 3.9;p=0.030) instead of '2.1 times the odds of a heroin-negative urine (95% CI 1.1 to 3.9;p=0.030)'.

We thank the reviewer for their comment and agree it would be better to report the odds ratios in a standard way. In the Abstract we have changed this to, "CM Attendance was superior to TAU in encouraging heroin abstinence. The odds of a heroin-negative urine in weeks 9-12 was statistically significantly greater in CM Attendance compared to TAU (OR=2.1; 95% CI:1.1 to 3.9,p=0.030)...". In the results we have changed this to, "Participants in the CM Attendance group had statistically significantly greater odds of a heroin-negative urine compared to those in the TAU group (OR=2.1; 95% CI 1.1 to 3.9;p=0.030)."

The reviewer commented, the authors claim 'CM Abstinence had 1.5 times the odds of attending fully as compared to TAU (CI 0.9 to 2.4;p=0.099)' but this crosses the point of no effect, which is 1. So how significant was this therefore. Even the p=0.099? This needs to be stated as such.

We have revised our wording in the reporting of this result and CM Abstinence compared to CM Attendance to be much clearer. We now state clearly that there was little difference. We have revised the wording in the Abstract to state. "CM Abstinence was not superior to TAU (OR = 1.6; 95% CI:0.85 to 3.01,p=1.46) or CM Attendance (OR=1.3;95% CI:0.68 -2.41,p=0.438) (no statistically significant difference) . In the results we have changed this to, "There were no statistically significant differences between CM Abstinence and TAU groups (OR=1.6; CI 0.9 to 3.0;p=0.146) and CM Attendance groups (OR =1.29; CI:0.68 to 2.41;p=0.438).."

The reviewer states that, The self-report outcomes are given as percentages, but without their 95% CI. They would have been better presented with their 95%CI.

The percentages included were intended to be descriptive, not inferential, and so we did not include their CIs. We have now included Table 5 with the self-reported heroin use statistics including Odds ratios and CIs in the manuscript.

Reviewer: 3

We thank the reviewer for their positive comments on the conduct of the trial, comprehensive reporting of results and relevance of our results for the adoption of future policies in the UK.

Abstract

The reviewer points out that the term “opioid use disorder” is repeatedly used to describe heroin users across the manuscript. While all heroin users have an OUD, not all people with OUD use heroin. The reviewer suggests we use “heroin use disorder” instead.

Thank you for this helpful suggestion. We have amended to “heroin use disorder” as suggested in the Title of the manuscript and in the text.

The reviewer states, Reading the abstract, it is unclear to me who are the participants exactly (i.e., adults with heroin use disorder). This should be specified.

We agree this was an omission and should be included. We have now added this to the abstract.

The reviewer states, It would be relevant to enumerate a few secondary outcomes as well.

We agree and have added to the methods section, Secondary outcomes include attendance, and self-reported drug use and physical and mental health.

The reviewer comments that, Across the manuscript, nonsignificant results are presented similarly to statistically significant results. In order to avoid misleading the reader, it would be important to be careful in the manner of presenting these results. For example, the authors could mention that this result is not significant, or even use the conditional tense (e.g., “CM Abstinence seemed to be superior to TAU, with an odds ratio of...).

We thank the reviewer for this sensible suggestion. We have now changed to include the reviewer’s suggestion that the group is either superior or not superior to TAU. In the Abstract we have changed this to, “ CM Attendance was superior to TAU in encouraging heroin abstinence. The odds of a heroin-negative urine in weeks 9-12 was statistically significantly greater in CM Attendance compared to TAU (OR=2.1; 95% CI:1.1 to 3.9,p=0.030)...” We have revised the wording in the Abstract to state. “ CM Abstinence was not superior to TAU (OR = 1.6; 95% CI:0.9 to 3.0,p=0.146) or CM Attendance (OR=1.3;95% CI:0.7 to 2.4,p=0.438) (not statistically significant differences)

The reviewer suggests that the conclusions should be more nuanced since (a) the reductions were not sustained in time, (b) the study has many important limitations, and (c) no improvement was observed regarding self-reported heroin use.

Thank you for these observations and the helpful recommendation. On reflection we agree and have amended the conclusions to: “ A pragmatically adapted CM intervention for routine use in UK drug services was moderately effective in encouraging heroin-abstinence compared with no CM only when targeted at attendance. CM targeted at abstinence was not effective.”

The Abstract has been edited to ensure that even with these revisions it has been kept to 300 words.

INTRODUCTION

The reviews states, the introduction is overall very concise and clear. However, it would be interesting if the authors provided more detail about the available evidence on CM efficacy in drug, opioid and heroin users.

We agree it would be helpful to have a little more detail. Thus we have added, "... including in treating drug use regardless of treatment setting, and for treating drug use (including cocaine, opiates and cocaine and poly-substance use) in opiate addiction treatment."

METHODS

The reviewer states, The study design is well described. However, the rationale justifying the utilization of clusters was not detailed in the article. Nevertheless, it was well explained in the previously published protocol. In order to meet the CONSORT 2010 item 2a, the authors should provide a bit more detail about that in the "Study design and setting" section.

Thank you for this comment. We agree it would be helpful to explain the rationale in this manuscript. Therefore we have added, "The unit of randomization was the drug clinic (cluster) rather than individual participant. This was for three reasons; 1) to reduce the likelihood of contamination if staff were delivering and patients receiving different interventions at the same drug clinic; 2) patients themselves constitute a local social network and individual randomization would be highly likely to encounter inter-service user contamination; and 3) participants receiving TAU would be denied an incentive offered to others in the same clinic, which might lead to low recruitment, poor compliance and high drop-out within this arm. Sites were recruited in stages and then randomized."

The reviewer asked, it would also be relevant to explain what the rationale is for the choice of the exclusion criteria.

We have amended the sentence on exclusion criteria to provide a rationale, "We excluded patients if they: could not read English and required an interpreter to understand a brief oral description of the study as they would be unable to understand the contingency management intervention provided; were pregnant or breastfeeding (due to being seen as special treatment population receiving special treatment provision); and/or were referred through the criminal justice pathway and were receiving a community sentence on condition of attending drug treatment as they would be subject to additional contingencies which might influence their behaviour."

The reviewers asks, If the information is available, for replicability purposes, it would be relevant to specify what the approximative duration of the weekly clinical appointments was.

We have added that the duration of clinical appointments ranged between 15 mins and 50 mins.

The reviewer reports, A major concern I have is about the UDS test that was used. What test was used exactly? How sensitive/specific is it? For how long following consumption can it detect heroin in urine? Unless I am mistaken, heroin can no longer be detected in urine after two days. Therefore, a weekly negative sample does not ensure that the patient is abstinent. While this design was chosen to

facilitate the ease of implementation, which seems reasonable, the limitations underlying this measurement should nevertheless be explicitly detailed. what were limitations of this ?

An instant UDS was undertaken using an individual, drug integrated cup test to detect opioids. Opioids can be detected between 1-3 days after use. We agree this will only indicate if a participant has used opioids for between one and three days before the test. While not providing confirmation of abstinence over the week period, we felt it did provide evidence that the participant was able to not use opioids for one to three days. We have added to the text, UDS were undertaken once a week using a drug integrated cup test to detect opioids. This test is able to detect opioids up to 1-3 days after use. Whilst, unable to confirm opioid abstinence over the week period, the test nevertheless, provided clinically significant evidence of the participants ability to abstain from using opioids over this briefer period.

The reviewer asked, Regarding the changes that were made to the protocol during the trial (first paragraph of page 7): please provide the rationale for these changes (CONSORT 2010 item 3b).

Thank you for pointing this out. We agree, we have not explained this very well . We have now added, After recruiting 13 clusters, with an attrition rate of 10%, larger than the expected 5%, clusters still recruiting were asked to increase recruitment from 20 to 22 participants.

RESULTS

The reviewer has stated, Please refer to comment #4, which is relevant for all the results section.

We have revised the results section to state, "Results from the analyses with and without imputation were similar (Table 3), so results from imputed data are discussed. Participants in the CM Attendance group had statistically, significantly greater odds of a heroin-negative urine at 9-12 weeks compared to those in the TAU group (OR=2.1; 95% CI 1.1 to 3.9;p=0.030). There were no statistically significant differences between the CM Abstinence and TAU (OR=1.6; CI 0.9 to 3.0;p=0.146) or CM Attendance groups (OR =1.3; CI:0.7 to 2.4;p=0.438). "

In addition, under attendance, we have revised the findings to state, "Participants in the CM Attendance group had statistically significantly greater odds of full attendance compared with those in TAU (OR=3.1; 95%CI 2.0 to 4.6;p<0.001). There were no statistically significant differences between CM Abstinence and TAU (OR=1.5; 95%CI 0.9 to 2.4;p=0.099)."

And again, amended to read, "Participants in the CM Abstinence group had statistically significantly greater risk of dropping out of the appointments before week 12 compared with those in TAU (Hazard Ratio (HR)=1.9;95% CI 1.5 to 2.5;p<0.001) and in CM Attendance)(HR= 1.7; 95% CI 1.2 to 2.3;p=0.002). (Figure 4)."

The reviewer has asked, more information should be provided as to why the initial objective of 660 participants was not met.

We have added to the text, "Although we recruited the target number of sites, we were unable to recruit 20 (and latterly 22) participants at all sites. Recruiting sites and participants at each site was challenging due to retendering of service contracts which affected many sites during the trial period. Two services were decommissioned during the trial."

The reviewer has asked, Figure 1: Why nine clinics were never approached?

We identified more potential clinics/sites than we needed. Nine were not approached as we achieved our target number.

The reviewer states, Table 1: There seem to be some differences between the groups. Statistical tests should be conducted to ensure that participants in the three interventions are similar, and statistical adjustments should be made if differences are found.

We thank the reviewer for their comment. However, we specified in our Statistical Analysis Plan that we wouldn't do this. It is generally accepted that statistical tests of baseline differences are not necessary in a randomised trial, where randomisation has been conducted properly any differences at baseline should be by definition due to chance. Also any statistical adjustment for baseline covariates should be pre-specified rather than post hoc (Altman D.G., Comparability of randomised groups. *The Statistician* 1985; 34:125-136, Assman S.F., Pocock S. J., Enos L. E., and Kasten L. E. Subgroup analysis and other (mis)uses of baseline data in clinical trials. *The Lancet*; Mar 25, 2000; 355, 9209) and we did not pre-specify adjustment for any baseline variables (except for performing ANCOVA analyses and moderation analyses).

The reviewer states, 16. Page 8 lines 51–52: if the two remaining participants were also using non-prescribed pharmaceutical heroin, this means that every single was using heroin in the month preceding the interview? The second sentence seems to be in contradiction with the first one. Please clarify.

We agree, this sentence is unclear. We have amended to, "All (552) self-reported using heroin in the month prior to interview (including two reporting using non-prescribed pharmaceutical heroin).

The reviewer states, Figure 3: It would be easier to follow and more coherent with the other figures if the x-axis was presenting weeks instead of days.

We thank the reviewer for their comment and agree. We have amended Figure 5 as suggested (attached)

The reviewer comments, The number of tables could be significantly reduced and it would be easier to interpret if the statistical results were presented with the associated descriptive statistics (e.g., table 2 with table 3, supp. table 3 with table 4 & 5, etc.).

Thank you for pointing this out. We agree and have removed Table 5. We have also moved the statistical results for self-reported heroin from supplementary material to the manuscript under the descriptive statistics. In the supplementary material, we have moved the statistical analysis for self-reported substance use under the descriptive data for substance use and the statistical analysis for health and social outcomes under descriptive data for these outcomes. In addition, we have re-named the table title to report "substance use", rather than non-opioid drug use, as includes alcohol and tobacco.

The reviewer reports, Heroin, crack, benzodiazepine, alcohol and cigarette use are reported. What about the use of other types of opioid? What is the rationale for not including them?

Other types of opioid were not reported as we were mainly concerned with patients/participants use of heroin as these were individuals with heroin-use disorder. Crack, benzodiazepines , alcohol and tobacco were the next most commonly reported substance categories seen in this patient group.

The reviewer comments that Supp. table 2 seems to be the same as table 5. If this is the case, it should be removed.

Thank you for pointing this out to us. Our apologies, this was an error. The table has now been removed from the supplementary data and left in the paper.

The reviewers state, Whenever the authors state that there is “no difference” between the groups (e.g., page 13 line 22), a statistical test should accompany that claim.

For clinical outcomes (health and social) we have inserted a sentence stating that these data can be found in the supplementary data. Please note crime was not a pre-specified secondary outcome and thus has been removed. Crime data 'will be presented', as intended, in the economic evaluation as stated in the methods section.

The reviewer comments, Page 13 line 39: The authors claim that the serious adverse events were not related to the trial intervention. How was that determined? What was considered to be a serious adverse event, specifically?

All SAE's were determined to be related or not to the trial intervention by the doctor at the relevant site, as usual for clinical trials. Serious Adverse events included those recommended for clinical trials ie. any AE that: results in death, is life-threatening (i.e. the individual was at immediate risk of death from the AE as it occurred), requires inpatient hospitalisation or prolongation of expected duration of hospitalization, results in persistent or significant disability/incapacity, is a congenital anomaly /birth defect, or is a medically important event or reaction. Medical judgement was exercised in deciding whether other situations should be considered serious events, such as important medical events that might not be immediately life threatening or result in death or hospitalisation but might jeopardise the patient or might require intervention to prevent one of the other outcomes listed above.

DISCUSSION

The reviewer states, Please refer to comment #10 which should also be addressed in this section.

We recognize this is a limitation and have added, “Opioids can only be detected up to 1-3 days after use so the UDS cannot confirm opioid abstinence over the week period, only provide evidence of abstinence over a few days, potentially reducing the impact of the reinforcer. “

The reviewer makes two final linked points. Firstly, the reviewer has stated, An important concern I have is that no effect was observed on self-reported heroin use. Considering that heroin use may not always be detected with a weekly UDS, this raises a major limitation. Indeed, the question is whether

it is possible that the only effect of CM on some patients is that they did not consume in the 1–2 days before their visit but were still consuming afterward. This important issue should be discussed in that section. Secondly, the reviewer also asked, In general, the authors should discuss more the limitations of the study and remain careful in their interpretations of the results, especially considering that these are not supported by the secondary outcomes (heroin use, functioning, wellbeing, depression, anxiety).

We agree with the first point that this is a limitation and should be discussed in the discussion. We have added, “ As CM Abstinence had no effect on either UDS or self-reported heroin use, it is possible that CM Abstinence only encouraged abstinence in the 1–3 days before an appointment and a UDS test.” We further add, in response to the second point: “While it may be argued that such behaviour change can be clinically significant to the individual patient, the absence of change in secondary outcomes suggests we should be cautious about inferring any significant clinical benefit.”

VERSION 2 – REVIEW

REVIEWER	Mao, Limin University of New South Wales, Centre for Social Research in Health
REVIEW RETURNED	06-Mar-2021
GENERAL COMMENTS	I think the authors have addressed all the major concerns raised by the three reviewers.
REVIEWER	Beaudoin, Mélissa University of Montreal, Psychiatry and addictology
REVIEW RETURNED	08-Mar-2021
GENERAL COMMENTS	I am satisfied with the authors' responses and modifications to the manuscript, and therefore I recommend its publication.